# RTify: Aligning Deep Neural Networks with Human Behavioral Decisions

**Yu-Ang Cheng**[*1], **Ivan Felipe Rodriguez**[*1], **Sixuan Chen**[1],
**Kohitij Kar**[2], **Takeo Watanabe**[1], **Thomas Serre**[1]
[1] Brown University    [2] York University
{yuang_cheng,ivan_felipe_rodriguez
sixuan_chen1,takeo_watanabe,thomas_serre}@brown.edu
k0h1t1j@yorku.ca

## Abstract

Current neural network models of primate vision focus on replicating overall levels of behavioral accuracy, often neglecting perceptual decisions' rich, dynamic nature. Here, we introduce a novel computational framework to model the dynamics of human behavioral choices by learning to align the temporal dynamics of a recurrent neural network (RNN) to human reaction times (RTs). We describe an approximation that allows us to constrain the number of time steps an RNN takes to solve a task with human RTs. The approach is extensively evaluated against various psychophysics experiments. We also show that the approximation can be used to optimize an "ideal-observer" RNN model to achieve an optimal tradeoff between speed and accuracy without human data. The resulting model is found to account well for human RT data. Finally, we use the approximation to train a deep learning implementation of the popular Wong-Wang decision-making model. The model is integrated with a convolutional neural network (CNN) model of visual processing and evaluated using both artificial and natural image stimuli. Overall, we present a novel framework that helps align current vision models with human behavior, bringing us closer to an integrated model of human vision.

## 1 Introduction

Categorizing visual stimuli is crucial for survival, and it requires an organism to make informed decisions in dynamic and noisy environments. This critical aspect of visual perception has driven the development of computational models to understand and replicate these processes. Traditionally, the field has followed two distinct paths.

On the one hand, image-computable vision models are used to predict behavioral decisions during (rapid) visual categorization tasks ranging from models of early- [1–3], mid- [4–6] and high-level vision [7–9] (see [10] for a review). More recently, these earlier models were superseded by deep convolutional neural networks (CNNs), which have become the de-facto choice for modeling behavioral decision [11–14]. Models are typically evaluated by estimating confidence scores computed for individual images, which are then correlated with similar scores derived for human observers(such as the proportion of correct human responses for each image). Such metrics ignore human reaction times (RTs); hence, current vision models only partially account for human decisions.

On the other hand, decision-making models have been used to explain how visual information gets integrated over time – predicting behavioral choices and RTs jointly. Notably, mathematical models, exemplified by evidence accumulation models such as the drift-diffusion [15–17] and linear ballistic

---

[*]These authors contributed equally to this work.

38th Conference on Neural Information Processing Systems (NeurIPS 2024).

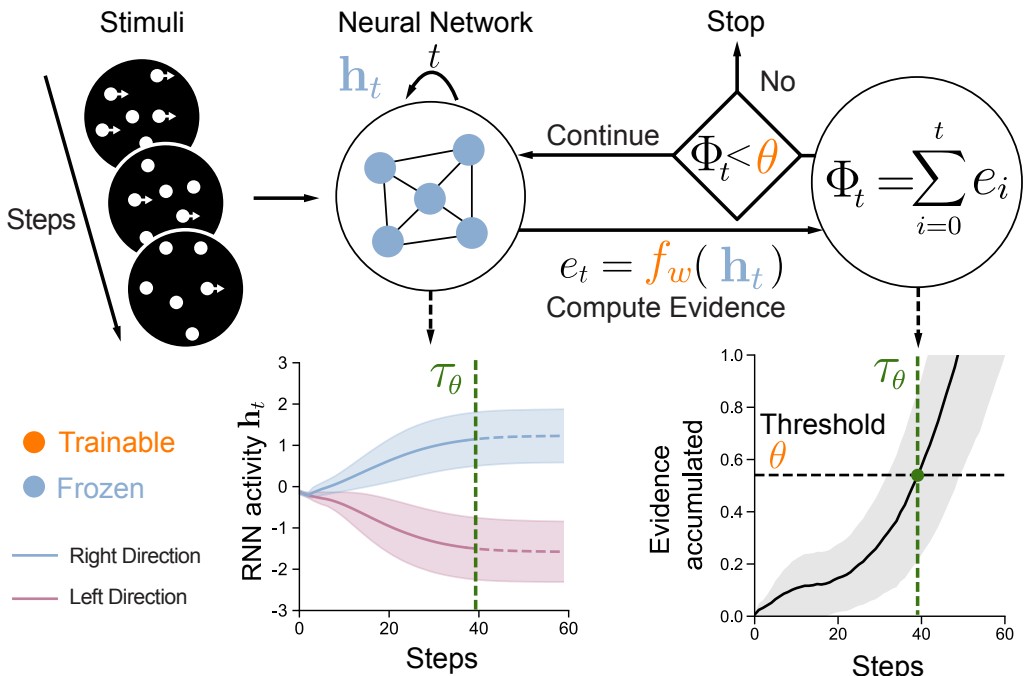

Figure 1: **Illustration of our RTify method.** The input is a visual stimulus represented by random moving dots, but the model can also accommodate color images and video sequences. We take a pretrained task-optimized RNN and use a trainable function $f_w$ to transform the activity of the network into a real-valued evidence measure, $e_t$, that will be integrated over time by an evidence accumulator, $\Phi_t$. When the evidence accumulator reaches the threshold $\theta$, processing stops, and a decision is taken. The time step at which the accumulated evidence passes this threshold $\tau_\theta$ is taken as the model RT for this stimulus.

accumulators [18, 19], have been quite successful in modeling an array of behavioral data (see [20] for a review). In addition, mechanistic models, including the Wong-Wang (WW) model, have provided insights into the underlying neural mechanisms [21, 22]. However, these efforts have primarily relied on traditional psychophysics tasks using simple, artificial stimuli, such as Gabor patterns [23] and random moving dots [24]. Beyond these easily parameterizable stimuli, these models have not been extended to deal with more complex, natural stimuli.

While both vision and decision-making models have contributed distinctively to our understanding of visual processes, a complete understanding of human vision will require their integration to explain the whole dynamics of visual decision-making. Recent neuroscience studies have leveraged recurrent neural network (RNN) models as the starting point for this integration [25] (see [26] for a review). More generally, recurrent processing has been shown to be necessary to account for both behavioral and neural recordings in object recognition tasks [27, 28].

Two promising approaches have been described that leverage RNNs to bridge the gap between decision-making and vision models [29, 30]. In [29], an RNN was trained to solve a visual classification task using the backpropagation-through-time (BPTT) learning algorithm. The entropy of the RNN outputs is computed at each timestep and used as a proxy for the network confidence. A decision is taken when the entropy reaches the threshold, halting further recurrent computation. In this approach, the recurrence steps are not differentiable, which prevents the use of gradient methods and inherently limits the complexity of the corresponding decision function. The resulting model may have difficulty predicting the entire distribution of RTs. Besides, the method is only applicable when human RTs are available because it requires an extensive search for the correct threshold value to fit human RT data.

An alternative approach was described in [30]. In [30], a convolutional RNN was trained on a visual classification task using contractor recurrent back-propagation (C-RBP). Besides, instead of searching for an optimal threshold to fit human RT data, a surrogate time-evolving 'uncertainty'

metric was estimated with evidential deep learning [31]. In this framework, model outputs are treated as parameters of a Dirichlet distribution, with the width of the distribution reflecting uncertainty. Remarkably, the resulting approach fits a range of experimental data well, without any supervision from human data. It is true that uncertainty and RTs are tightly coupled and are both affected by task difficulty. However, uncertainty and RTs are conceptually different. Experimental results show that uncertainty and RTs can be positively or negatively correlated [32], and even double dissociated [33] under different experimental conditions. Therefore, a good model of uncertainty is not guaranteed to be a good model of RTs.

Here, based on extensive cognitive neuroscience research [34–37], we introduce a novel trainable module called RTify to allow an RNN to dynamically and nonlinearly accumulate evidence. First, this module can be trained to learn to make human-like decisions using direct human RT supervision. Our results suggest that incorporating a dynamic evidence accumulation process, compared to the entropy heuristics used in [29], can help better capture human RTs. Second, we show how the same general approach can also be used to train an RNN to learn to solve a task with an optimal number of time steps via self-penalty. Our results show that human-like RTs naturally emerge from such ideal-observer models without explicit supervision from human RT data. Hence, our framework is general enough to allow the fitting of human RT data as done in [29] and/or the training of a neural architecture that can optimally trade speed and accuracy via self-penalty as done in [30].

**Contributions:** Overall, our work makes the following contributions: **(i)** We present RTify, a novel computational approach to optimize the recurrence steps of RNNs to account for human RTs. This enables dynamic nonlinear evidence accumulation learned through back-propagation (a) to fit human data or (b) optimally balance speed and accuracy. **(ii)** We comprehensively demonstrate the effectiveness of our framework for modeling human RTs across a diverse range of psychophysics tasks and stimuli. Our method consistently outperforms alternatives with and without explicit training on human data. **(iii)** As an illustrative example of the framework's potential, we extend the WW decision-making model [22] to create a biologically plausible, multi-class compatible, and fully differentiable RNN module. We show that the enhanced neural circuit can be used as a drop-in module for CNNs to fit human RTs.

## 2   RTify: Overview of the method

First, we explain how our RTify module is applied to a pre-trained RNN. Then, we will explain how to tune a deep-learning RNN-based implementation of the WW model to RTify feedforward networks.

We start with a task-optimized RNN with hidden state $\mathbf{h}_t$, which remains frozen. We then train a learnable mapping function $f_w : \mathbf{R}^k \to \mathbf{R}$ that summarizes the state of the neural population at each time step $t$ by mapping the RNN hidden state $\mathbf{h}_t$ to some "evidence": $e_t = f_w(\mathbf{h}_t)$. At every time step, the evidence is integrated via an "evidence accumulator" $\Phi_t = \sum_{i=1}^{t} e_i$, and when the accumulated evidence passes a learnable threshold $\theta$, the model is read out, and a decision is made. The time step at which the accumulated evidence first passes this threshold is given by $\tau_\theta(\Phi) = \min\{t : \Phi_t > \theta\}$, and is treated as model RTs. In summary, $\tau_\theta(\Phi)$ is directly influenced by the threshold $\theta$ and by $w$ through $\Phi_t$.

To align the model RTs with human RTs, or to penalize the model for excessive time steps, we need to optimize a loss function over $\tau_\theta(\Phi)$. In the most general case, we first consider $F(\tau_\theta(\Phi))$ as our loss function to illustrate how we approximate its gradient. Since our goal is to minimize $F$, we will need to calculate the gradient $\frac{\partial F(\tau_\theta(\Phi))}{\partial \theta}$ and $\frac{\partial F(\tau_\theta(\Phi))}{\partial w}$. Following the chain rule, we get

$$\frac{\partial F(\tau_\theta(\Phi))}{\partial w} = \frac{\partial F(\tau_\theta(\Phi))}{\partial \tau_\theta(\Phi)} \cdot \frac{\partial \tau_\theta(\Phi)}{\partial w}, \frac{\partial F(\tau_\theta(\Phi))}{\partial \theta} = \frac{\partial F(\tau_\theta(\Phi))}{\partial \tau_\theta(\Phi)} \cdot \frac{\partial \tau_\theta(\Phi)}{\partial \theta}. \tag{1}$$

This means a crucial step involves estimating the gradient of $\tau_\theta(\Phi)$ over the trainable parameters $w$ and $\theta$ of the RTify.

The primary challenge that arises when extracting the gradient of $\tau_\theta(\Phi)$ over the trainable parameters $w$ and $\theta$ is the non-differentiability of $\tau_\theta(\Phi)$, which prevents the direct use of the backpropagation algorithm. This is because $\tau_\theta(\Phi)$ lies in the integer space and requires non-differentiable operations

such as the minimum function and the inequality. We will relax the original formulation to circumvent this issue, enabling us to approximate the gradient.

Assume that $\Phi(t)$ is a continuous function on the closed interval $[1, N]$, representing the accumulation of evidence over time. Define $\tau_\theta(\Phi) = \min\{t \in [1, N] : \Phi(t) > \theta\}$, which is the earliest time when $\Phi(t)$ exceeds the threshold $\theta$. We can use a first-order Taylor expansion to find the following approximation:

$$\frac{\partial \tau_\theta}{\partial w} \approx \frac{\partial \tau_\theta^*}{\partial w} \approx \frac{1}{\Phi_t - \Phi_{t-1}} \cdot \left(-\frac{\partial \Phi_t}{\partial w}\right), \frac{\partial \tau_\theta}{\partial \theta} \approx \frac{\partial \tau_\theta^*}{\partial \theta} \approx \frac{1}{\Phi_t - \Phi_{t-1}} \tag{2}$$

Fig. S1 provides a visual intuition for the approximation, and the full derivation can be found in the SI A.1. This leads to the following gradients for our trainable parameters:

$$\frac{\partial F(\tau_\theta(\Phi))}{\partial w} \approx \frac{\partial F(\tau_\theta(\Phi))}{\partial \tau_\theta(\Phi)} \cdot \frac{1}{\Phi_t - \Phi_t} \cdot \left(-\frac{\partial \Phi_t}{\partial w}\right), \frac{\partial F(\tau_\theta(\Phi))}{\partial \theta} \approx \frac{\partial F(\tau_\theta(\Phi))}{\partial \tau_\theta(\Phi)} \cdot \frac{1}{\Phi_t - \Phi_{t-1}}. \tag{3}$$

Since $F$ and $\Phi$ are both differentiable by nature, the gradients in Eqs. 2 and 3 are all computable after this approximation.

Under this framework, we consider two different scenarios: Training with human data directly ("supervised"; see Section 2.1) or training with "self-penalty", which involves no explicit human data but uses a penalty term that spontaneously leads to decision times similar to human RTs (e.g., achieving an optimal speed-accuracy trade-off; see Section 2.2).

## 2.1 Predicting human decisions with direct "supervision"

Human behavioral decisions in the random dot motion (RDM) tasks used in decision-making studies [16, 24, 38] are typically summarized as histograms similar to those shown in Fig. 2. Here, histograms are computed for RTs for correct and incorrect trials corresponding to individual experimental conditions (such as coherence levels shown here; see section 3 for details). Moreover, RTs for incorrect trials are turned into negative RTs. Combined with correct RTs (which stay positive), one single histogram is used for capturing both accuracy (the proportion of positive values) and RTs. To measure the goodness of fit between human RTs and model RTs, we use a mean squared error loss (MSE) between histograms of model RTs and human RTs.

In the object recognition task [39], only RTs averaged across all participants were available. We can match human data on a stimulus-by-stimulus basis using the negative correlation loss between model and human RTs.

## 2.2 Predicting human decisions with "self-penalty"

Our framework allows us to develop an "ideal-observer" RNN model explicitly trained to balance the computational time required for solving a particular classification task and its own accuracy for the task, i.e., a speed-accuracy trade-off.

To achieve this, we add a regularizer to the cross-entropy loss to encourage the RNN to jointly maximize task accuracy while minimizing the computational time needed to solve the task. With $l$ as the output logits of the network, $\hat{y}$ as the output probabilities of the network, and $y$ as the ground truth, we can write our penalty term for a single sample as:

$$L_{\text{self-penalized}} = L_{CCE}(y, \hat{y}) + \lambda\left(\mathbf{l_y} \cdot \tau_\theta\right) \tag{4}$$

where $\lambda$ is a hyperparameter for controlling the strength of the penalty, $\mathbf{l_y}$ refers to the logit value of the correct label, $\tau_\theta$ is the model decision time. This penalty means that the model will be penalized for using too much time, especially for higher confidence (higher $\mathbf{l_y}$).

## 2.3 RTifying feedforward networks

To integrate temporal dynamics into feedforward neural networks (e.g., CNNs), we describe an RNN module that approximates the WW neural circuit model [22]. The original WW model is a biophysically-realistic neural circuit model of two-alternative forced choices via the temporal

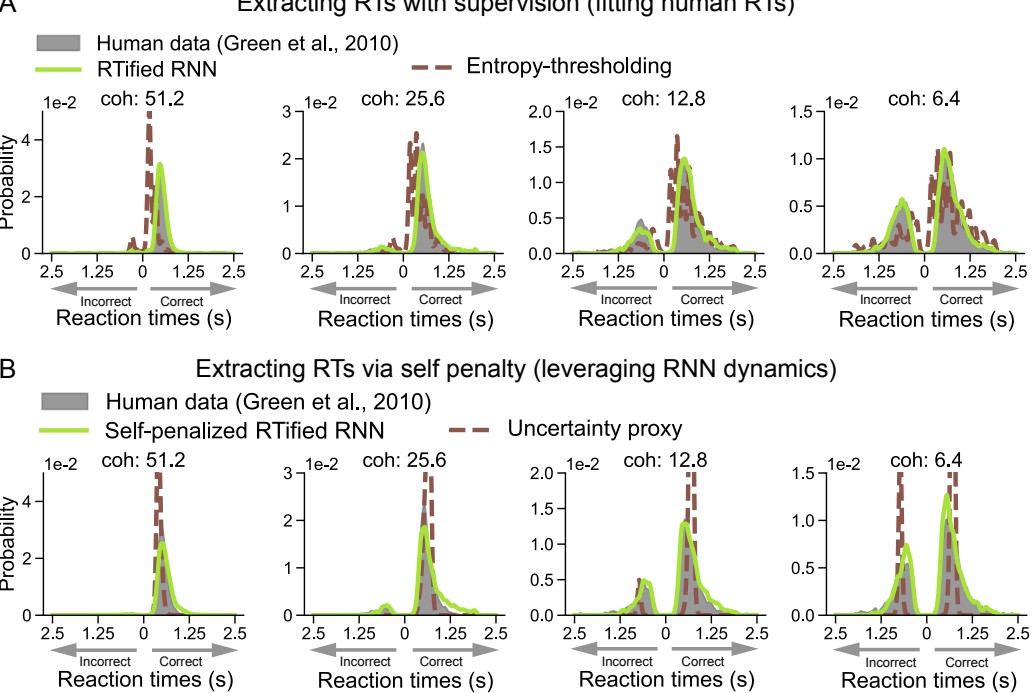

Figure 2: **RTified model evaluation on a RDM task [24]**. Human data are shown as a gray shaded area, and model fits are shown for **(A)** the "supervised" setting where human behavioral responses are used to train the models and **(B)** the "self-penalized" setting where no human data is used. Our approach (green) outperforms the two alternative approaches (brown), i.e., entropy-thresholding [29] for the "supervised" and uncertainty proxy [30] for the "self-penalized" settings (see Fig. 4 for MSE comparisons and Fig. S3 for all coherences).

accumulation of sensory evidence in two distinct neural populations. It takes a constant scalar input (representing a stimulus parameter such as the degree of coherence for randomly moving dot stimuli). It outputs an RT when the activity of either population reaches a decision threshold.

However, the original WW model has limitations: its parameters must be manually tuned to fit observed data, and it is restricted to binary classification tasks with simple artificial parametric stimuli (e.g., Gabor patterns). To overcome these limitations, we extend the WW model. First, we replace the scalar input with a feedforward neural network (e.g., a CNN), enabling the model to process complex stimuli such as natural images. Second, we generalize the model from two populations to $M$ populations, effectively increasing the number of neural populations to handle multi-class classification problems. Third, we RTify the model to make all parameters trainable via backpropagation. This allows the model to automatically learn the optimal parameters to fit human RTs (see Fig. 3 for an illustration of the multi-population case and SI Fig. A.2 for an illustration of the two-population case).

## 3    Experiments

In this section, we validate our RTify framework on two psychophysics datasets: the RDM dataset [24, 38] and a natural image categorization dataset [39]. As a side note, all models were trained on single Nvidia RTX GPUs (Titan/3090/A6000) with 24/24/48GB of memory each. All training can be completed in approximately 48 hours. Code and data are available at `https://github.com/Yu-AngCheng/RTify`.

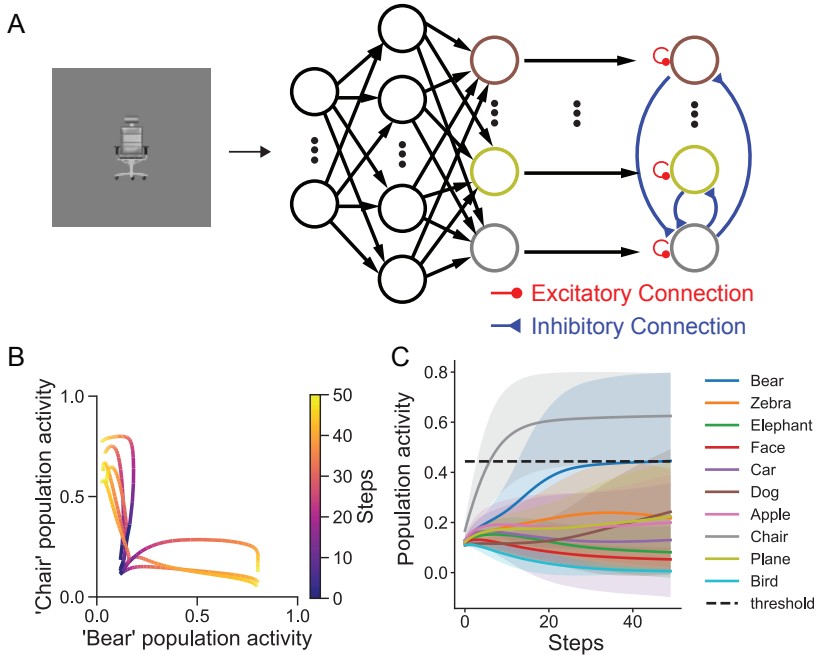

Figure 3: **Illustration of RTifying feedforward neural networks**. We develop a multi-class compatible and fully differentiable RNN module based on the WW model [21, 22]. This module is implemented as an attractor-based RNN, and is stacked on top of a feedforward neural network. The feedforward neural network first takes an image as the input. Outputs from classification units of the network are then sent to RTified WW (**A**). Information is accumulated by multiple populations of neurons in RTified WW while they compete with each other (**B**). A decision is made and the process stops when one of the populations reaches a threshold. The number of time steps needed for the RTified WW to reach the threshold is used to predict human RT (**C**).

## 3.1 Random dot motion task

The RDM task is a classic experimental paradigm used to test temporal integration that has been extensively used in psychophysics [40], human imaging [41], and electrophysiology studies [42]. The stimuli in this task consist of dots moving on a screen toward a predefined direction vs. randomly. For each time step, each dot only has a specific probability (coherence) $(0.8\%, 1.6\%, 3.2\%, 6.4\%, 12.8\%, 25.6\%,$ or $51.2\%)$ to move towards the pre-defined direction, making the task non-trivial. The participants must integrate motion information across time and report it when they are sufficiently confident. The original experimental data are from [24, 38], where 21 young adult participants performed around 40,000 trials in total over 4 consecutive days.

First, we trained an RNN consisting of 5 convolutional blocks (Convolution, BatchNorm, ReLU, Max pooling) and a 4096-unit LSTM with BPTT. In the original experiment, the stimuli were shown on a 75 Hz CRT monitor for up to 2 seconds, and therefore, we also trained our RNN for the RDM stimuli for 150 frames. The RNN was trained for 100 epochs using the Adam optimizer with a learning rate of 1e-4 at full coherence (c = 99.9%) for the first 10 epochs as a warm-up and 1e-5 at all coherence levels for the remaining 90 epochs. Next, we trained our two different RTify modules. For fitting human RTs, it was trained for 10,000 epochs, and for self-penalty, it was trained for 20,000 epochs. In both cases, the Adam optimizer were used, and the weights of the task-optimized RNN were frozen while training the RTify modules.

We trained the first RTify module to predict human RTs by fitting human RT distributions. Here, positive RTs refer to RT with correct choices, and negative RTs refer to RTs with incorrect choices. Therefore, one distribution incorporates both speed and accuracy information from behavioral choices. Results are shown in Fig. 2. Our RTify model can predict the full RT distribution across all coherence levels. In comparison, the entropy-thresholding approach by [29] fails to capture the full distribution (see Fig. 2 for coherence = 51.2% to 6.4% and Fig. S3 for all coherences). Importantly, our method

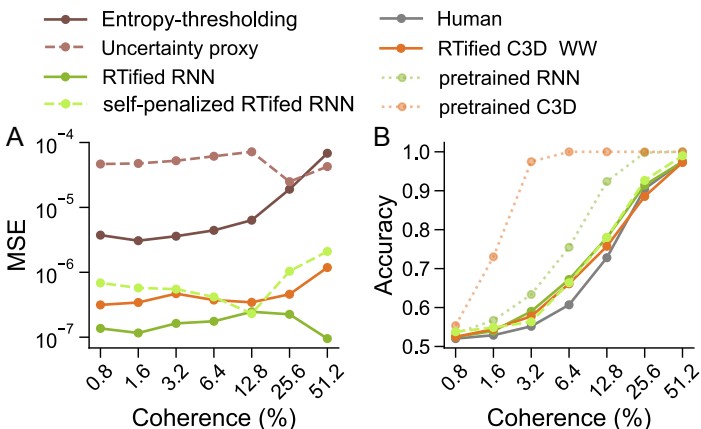

Figure 4: **(A) MSE comparisons for the RDM task [24] for all coherence levels.** The RTified model trained in the "supervised" setting (i.e., with human behavioral responses; green solid line) performs better (lower MSE) than entropy-thresholding [29] (brown solid line) under all coherence levels. Similarly, the RTified model trained in the "self-penalized" setting (i.e., without human data; green dash line) performs better than uncertainty proxy [30] (brown dash line). With the help of our RTified WW module (orange solid line), a convolution neural network (C3D) can also fit the data better than entropy-thresholding [29]. **(B) Classification accuracy comparisons between pretrained and RTified models for the RDM task [24].** The RTified model trained with human RTs data in the "supervised" setting (green solid line) and in the "self-penalized" setting (green dash line) achieve human-like classification accuracy under all coherence levels compared with the pretrained model without RTify (green dotted line). With the help of our RTified WW module (orange solid line), a CNN (C3D) matches human accuracy better than the pretrained model without RTify (orange dotted line).

surpasses entropy-thresholding approach [29] (two-sided Wilcoxon signed-rank test, $p < .05$; for MSE comparisons, see Fig. 4).

It is well known that there is a trade-off between RTs and accuracy in cognitive tasks. To investigate this relationship further, we extended our analysis to examine whether the model's accuracy would approximate human accuracy when it was fitted solely on human RTs without using human accuracy data. Given that conventional RT distributions encompass both RTs and accuracy information, we restricted our fitting procedure to the positive part of the distribution. That is, RTs corresponding to correct responses to prevent inadvertently incorporating human accuracy into the model. Remarkably, by fitting the model on human RTs only, the model naturally reached a classification accuracy comparable to human performance (see Fig. 4).

We also trained a self-penalized RTify module without any human data. This ideal-observer RNN model was trained to minimize the time steps needed for solving the RDM task (see section 2.2 for details). Human-like RTs emerge naturally in this neural network (See Fig. 2). Our model predicts RT data much better than previous approaches, which use a measure of uncertainty computed over the RNN as a surrogate metric. As can be seen, the resulting model tends to overfit the modes of the distribution.(see Fig. 2 for coherence = 51.2% to 6.4% and Fig. S3 for all coherences). Our method surpasses uncertainty proxy approach [30] (two-sided Wilcoxon signed-rank test, $p < .05$, for MSE comparison, see Fig. 4). We also checked the model classification accuracy before and after self-penalized RTify. Interestingly, we also found the self-penalized RTified RNN demonstrated a human-like classification accuracy (see Fig. 4).

### 3.2 Object recognition task

Unlike previous visual decision-making models, we want to show that our method can also be applied to natural images and multi-class datasets. Specifically, we consider an object recognition task, a classic paradigm used extensively in computer vision [43, 44] and cognitive neuroscience studies [45, 46]. The original data is from [39], where 88 participants perform the task through Mturk. The stimuli in this task belong to 10 categories, and for each category, there are 20 natural images

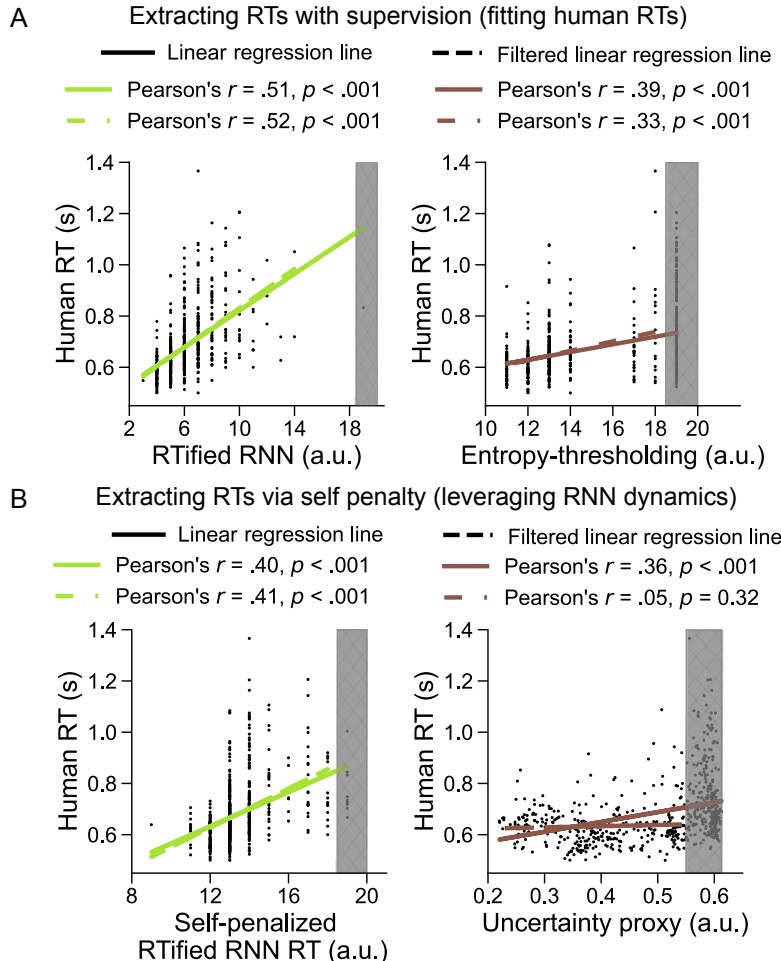

Figure 5: **RTified model evaluation on an object categorization task [39].** Model vs. human RT predictions for our RTified model (green) vs. alternative approaches (brown) **(A)** in the "supervised" setting where human behavioral responses are used to train the model and **(B)** the "self-penalized" setting where no human data is used. Solid lines are linear regression fits between model and human RTs. Crossed-shaded areas and the dashed lines are controls to show the fits after removing the highest model RTs. Our approach outperforms the two alternative approaches, i.e., entropy-thresholding [29] for the "supervised" setting and uncertainty proxy [30] for the "self-penalized" setting.

taken from the COCO dataset [47] and 112 synthetically generated images with different backgrounds and object positions.

We first train our RNN with BPTT to perform a 10-way classification task. In the original study, participants performed a binary classification task. However, since the individual binary pairs were not saved, we trained the model in a 10-class classification task. We used Cornet-S [48] pretrained on the Imagenet dataset [43] because it was used in the original study and it achieves a relatively high brainscore in terms of explaining neural activities [49]. We trained the network for 100 epochs, using the Adam optimizer with a learning rate of 1e-4 and a learning scheduler (StepLR) with a step size of 2,000. We used 20 timesteps (instead of 2 in the original model) for the IT layer in the Cornet to achieve high temporal resolution. Results show that our RNN achieves 75.2% on a held-out test set. Similarly, we trained our two different RTify modules. For fitting human RTs, it was trained for 100,000 epochs, while for self-penalty, it was trained for 10,000 epochs with a learning scheduler (StepLR) with a step size of 2,000 and a gamma of 0.3. In both cases, Adam optimizers were used, and the weights of the task-optimized RNN were frozen while training the RTify modules.

We then extracted RTs from our RNN to fit human RTs. Notably, model RTs significantly correlate with the human RT observed in the psychophysics experiment ($r = .51, p < .001$). Besides, our

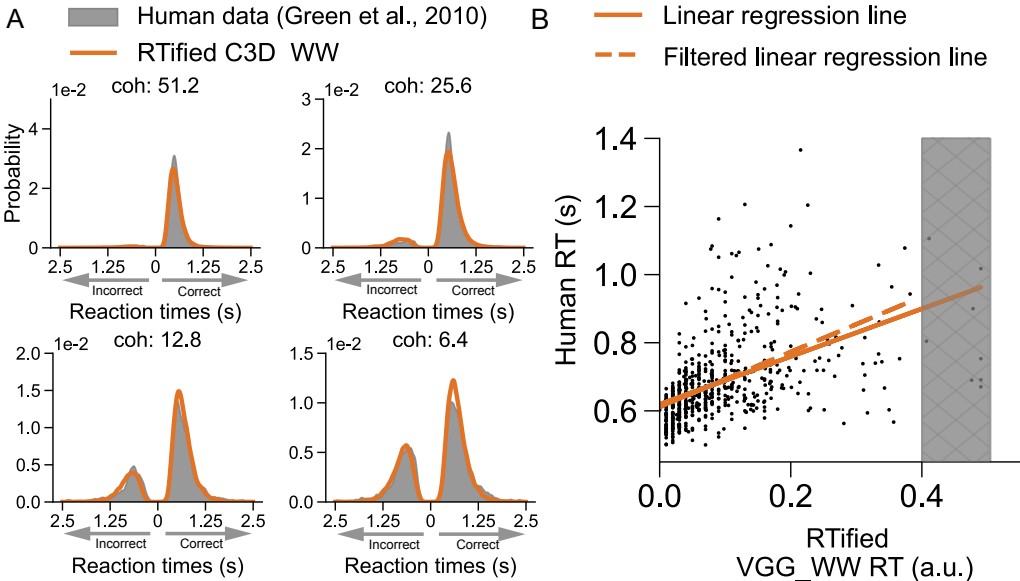

Figure 6: **RTified WW model evaluation.** We combine our RTified WW module with **(A)** a 3D CNN to fit human RTs collected in an RDM task [24] (see Fig. 4 for MSE comparisons with other methods) and **(B)** a VGG to fit human RTs in a rapid object categorization task [39] (Crossed-shaded areas and the dashed lines are controls to show the fits after removing the highest model RTs).

method also surpasses entropy-thresholding [29] (bootstrapping shows that our method is superior to theirs with a probability of 99.9%). We also extract RTs from an ideal-observer RNN model trained with a time self-penalty (see section 2.2 for details). We show here that model RTs are significantly correlated with human RTs (see Fig. 5, $r = .40, p < .001$). This failed to be captured by uncertainty proxy [30]. We argue this in two ways. First, bootstrapping shows that our method is superior to theirs with a probability of 87.1%. Second, and most importantly, although the uncertainty seems to correlate with human RTs in the dataset ($r = .36, p < .001$), it only applies to high-uncertainty cases. When trials with high model uncertainty are excluded, uncertainty shows no significant correlation with human RTs ($r = .05, p = 0.32$). However, using our method, the correlation remains strong and significant ($r = .41, p < .001$).

## 3.3 RTifying feedforward neural networks

Given the prevalence of feedforward networks (e.g. CNNs) and their incredible performance in visual tasks, a natural question is how to align such networks in the temporal domain of decision-making. We thus developed a biologically plausible, multi-class, differentiable RNN module based on the WW recurrent circuit model [21, 22]. This module can be stacked on top of any neural network, even if not recurrent, and can be used to align model RTs with human RTs.

For the RDM task, we take a 3D CNN with 6 convolutional blocks (convolution, BatchNorm, ReLU, Max pooling) and an MLP. We train the network for 100 epochs using the Adam optimizer with a learning rate of 1e-4 at full coherence (c = 99.9%) for the first 10 epochs as a warm-up and 1e-5 at all coherence levels for the remaining 90 epochs. Since this model is not an RNN, it has no temporal dynamics. Therefore, we drop in our WW module and further train the WW module for 5,000 epochs using the Adam optimizer with a learning rate of 1e-4, a StepLR scheduler with a step size of 1,000 and gamma of 0.3, and a grad clip at 1e-5. Interestingly, when we RTify the C3D model using the WW module, it is able to capture the distribution of human RTs across all coherence levels, see Fig. 6A for coherence = 51.2% to 6.4% and Fig. S4 for all coherences, for MSE comparison see Fig. 4). Furthermore, we also observed a human-like classification accuracy for the model when it is solely trained to fit human RTs but not human accuracy (see Fig. 4).

Similarly, we take a VGG-19 pre-trained on Imagenet for the object recognition task and fine-tune it on the dataset provided by Kar et al. [39] in a 10-class classification way for the abovementioned reasons. We train the model for 100 epochs using a batch size of 32. The optimizer was AdamW,

with a learning rate of 1e-5. We use a OneCycleLR scheduler adjusted after 10 epochs of warm-up. Results show that our RNN achieves 81.6% on a held-out test set. We further train the WW module for 100,000 epochs using the Adam optimizer with a learning rate of 1e-4 and a grad clip at 0.0001. By using the multi-class version of the WW model, we show that the RTify VGG also exhibited a significant correlation with human data ($r = .49, p < .001$, see Fig. 6B).

## 4    Conclusion

We have described a computational framework to train RNNs to learn to dynamically accumulate evidence nonlinearly so that decisions can be made based on a variable number of time steps to approximate human behavioral choices, including both decisions and RTs. We showed that such optimization can be used to fit an RNN directly to human behavioral responses. We also showed that such a framework can be extended to an ideal-observer model whereby the RNN is trained without human data but with self-penalty that encourages the network to make a decision as quickly as possible. Under this setting, human-like behavioral responses naturally emerge from the RNN – consistent with the hypothesis that humans achieve a speed-accuracy trade-off. Finally, we provided an RNN implementation of a popular neural circuit decision-making model, the WW model, as a trainable deep learning module that can be combined with any vision architecture to fit human behavioral responses. Our computational framework provides a way forward to integrating image-computable models with decision-making models, advancing toward a more comprehensive understanding of the brain mechanisms underlying dynamic vision.

**Limitations**    Certain limitations will need to be addressed in future work. Most of the human data used in our study remains relatively small-scale and is limited primarily to synthetic images because more naturalistic benchmarks only include behavioral choices [44, 50] and lack RT data. To properly evaluate our approach and that of others, larger-scale psychophysics datasets using more realistic visual stimuli will be needed. There is already evidence that large-scale psychophysics data can be used to effectively align AI models with humans [51, 52]. We hope this work will encourage researchers to collect novel internet-scale benchmarks that include both behavioral choices and RTs.

**Broader Impacts**    As AI vision models become more prevalent in our daily lives, ensuring their trustworthy behavior is increasingly important [53, 54]. Our framework contributes to this effort by exploring how to align certain aspects of models' behavior with human responses in specific contexts. While our approach is limited to predicting RT distributions, it constitutes a first step toward more human-aligned AI models.

## Acknowledgments and Disclosure of Funding

This work was supported by NSF (IIS-2402875), ONR (N00014-24-1-2026) and the ANR-3IA Artificial and Natural Intelligence Toulouse Institute (ANR-19-PI3A-0004) to T.S and National Institutes of Health (NIH R01EY019466 and R01EY027841) to T.W. Computing hardware was supported by NIH Office of the Director grant (S10OD025181) via Brown's Center for Computation and Visualization (CCV).

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

# A  Appendix / Supplemental material

## A.1  Complete derivation of the differentiable framework

**Proposition 1.** *Let us define $\tau_\theta^*(\Phi_t) = \min\{t \in \mathbb{R}_{[1,N]} : \Phi_t > \theta\}$ as the time in which $\Phi_t$ reaches the threshold of activity $\theta$. Provided that $\Phi_t$ is continuously differentiable:*

$$\frac{\partial \tau_\theta^*}{\partial w} \approx \frac{1}{\Phi_t - \Phi_{t-1}} \cdot (-\frac{\partial \Phi_t}{\partial w}) \tag{5}$$

*Proof.* By definition,

$$\tau_\theta^*(\Phi_t) = \min\{t \in \mathbb{R}_{[1,N]} : \Phi_t > \theta\}.$$

Let us consider a small change $\delta\Phi$. By the Taylor expansion, we can write:

$$\tau_\theta^*(\Phi_t + \delta\Phi) \approx \tau_\theta^*(\Phi_t) + \frac{\partial \tau_\theta^*}{\partial \Phi_t} \delta\Phi.$$

On the other hand, $t$ is considered a continuous value. By our definition, we can induce a small change in the value of $\Phi_t$ by introducing a small change in time $\delta\Phi$, which we can write in the following way:

$$\Phi_{t+\delta t} \approx \Phi_t + \frac{\partial \Phi_t}{\partial t} \delta t$$

Without loss of generality, let us take $\delta t = \left(\frac{\partial \Phi_t}{\partial t}\right)^{-1} \delta\Phi$ and then

$$\Phi_{t+\delta t} \approx \Phi_t + \frac{\partial \Phi_t}{\partial t} \left(\frac{\partial \Phi_t}{\partial t}\right)^{-1} \delta\Phi = \Phi_t + \delta\Phi$$

Now,

$$\tau_\theta^*(\Phi_t + \delta\Phi) = \min\{t \in \mathbb{R}_{[1,N]} : \Phi_t + \delta\Phi > \theta\}$$
$$\approx \min\{t \in \mathbb{R}_{[1,N]} : \Phi_{t+\delta t} > \theta\}$$
$$= \min\{t \in \mathbb{R}_{[1,N]} : \Phi_{t+\left(\frac{\partial \Phi_t}{\partial t}\right)^{-1}\delta\Phi} > \theta\}$$

Note that, by definition, if the evidence is evolving by $\Phi_{t+\Delta}$. It just means the whole function is moving along the t-axis, and therefore the minimum time to pass the threshold would be given by $\tau_\theta^* - \Delta$, therefore

$$\min\{t \in \mathbb{R}_{[1,N]} : \Phi_{t+\left(\frac{\partial \Phi}{\partial t}\right)^{-1}\delta\Phi}\} = \min\{t \in \mathbb{R}_{[1,N]} : \Phi_t\} - \left(\frac{\partial \Phi_t}{\partial t}\right)^{-1} \delta\Phi$$

$$= \tau_\theta^*(\Phi) - \left(\frac{\partial \Phi}{\partial t}\right)^{-1} \delta\Phi$$

Finally, we can join the two sides and obtain:

$$\tau_\theta^*(\Phi_t + \delta\Phi) \approx \tau_\theta^*(\Phi) + \frac{\partial \tau_\theta^*}{\partial \Phi} \delta\Phi = \tau_\theta^*(\Phi) - \left(\frac{\partial \Phi}{\partial t}\right)^{-1} \delta\Phi$$

From this, we can deduce:

$$\frac{\partial \tau_\theta^*}{\partial \Phi} = -\left(\frac{\partial \Phi}{\partial t}\right)^{-1} \tag{6}$$

$$\frac{\partial \tau_\theta^*}{\partial \Phi_t} \approx -\left(\frac{\Phi_t - \Phi_{t-1}}{t - (t-1)}\right)^{-1} \tag{7}$$

$$\frac{\partial \tau_\theta^*}{\partial w} \approx -\frac{1}{\Phi_t - \Phi_{t-1}} \frac{\partial \Phi_t}{\partial w} \tag{8}$$

So the first part of Eq 2 in the main text is proved. □

Similarly, we can derive $\frac{\partial \tau_\theta^*}{\partial \theta} \approx \frac{1}{\Phi_t - \Phi_{t-1}}$.

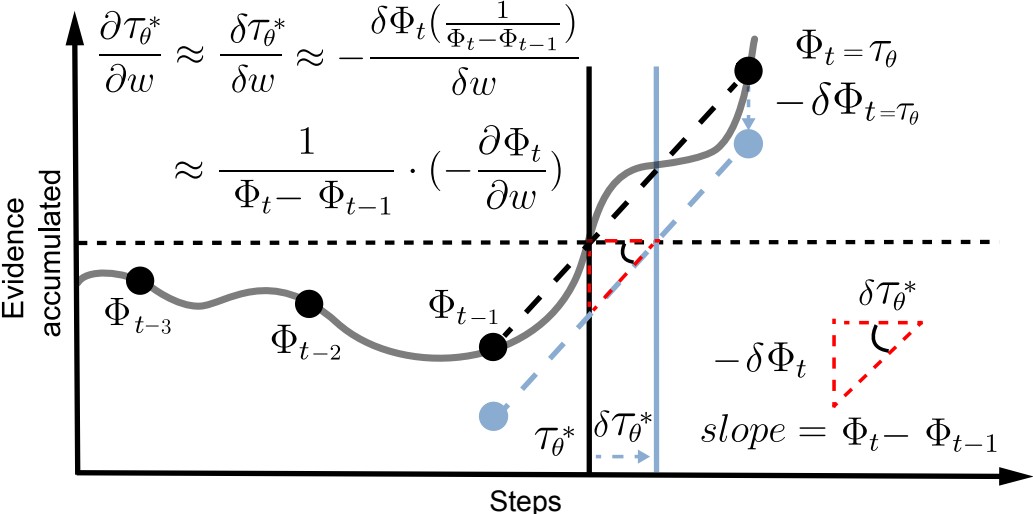

Figure S1: **Illustration of Mathematics Proof** The discrete RNN step $\tau_\theta$ is not differentiable. Therefore, we introduce a piecewise linear approximation of the accumulated evidence over time $\Phi_t$. Consider the effect of changes in $\Phi_t$ on $\tau_\theta(\Phi)$. A small perturbation in $\Phi_t$ will produce a proportional change in the time it takes for the accumulated evidence to cross the threshold $\theta$, thereby inducing a shift in time. In simple terms, fine-tuning $w$ to decrease $\Phi_t$ will delay the time at which the network crosses the threshold $\theta$ thus increasing $\tau_\theta(\Phi)$, while fine-tuning $w$ to increase $\Phi_t$ will cause the threshold to be crossed earlier thus decreasing $\tau_\theta(\Phi)$.

**A.2 Illustration of the RTified WW circuit**

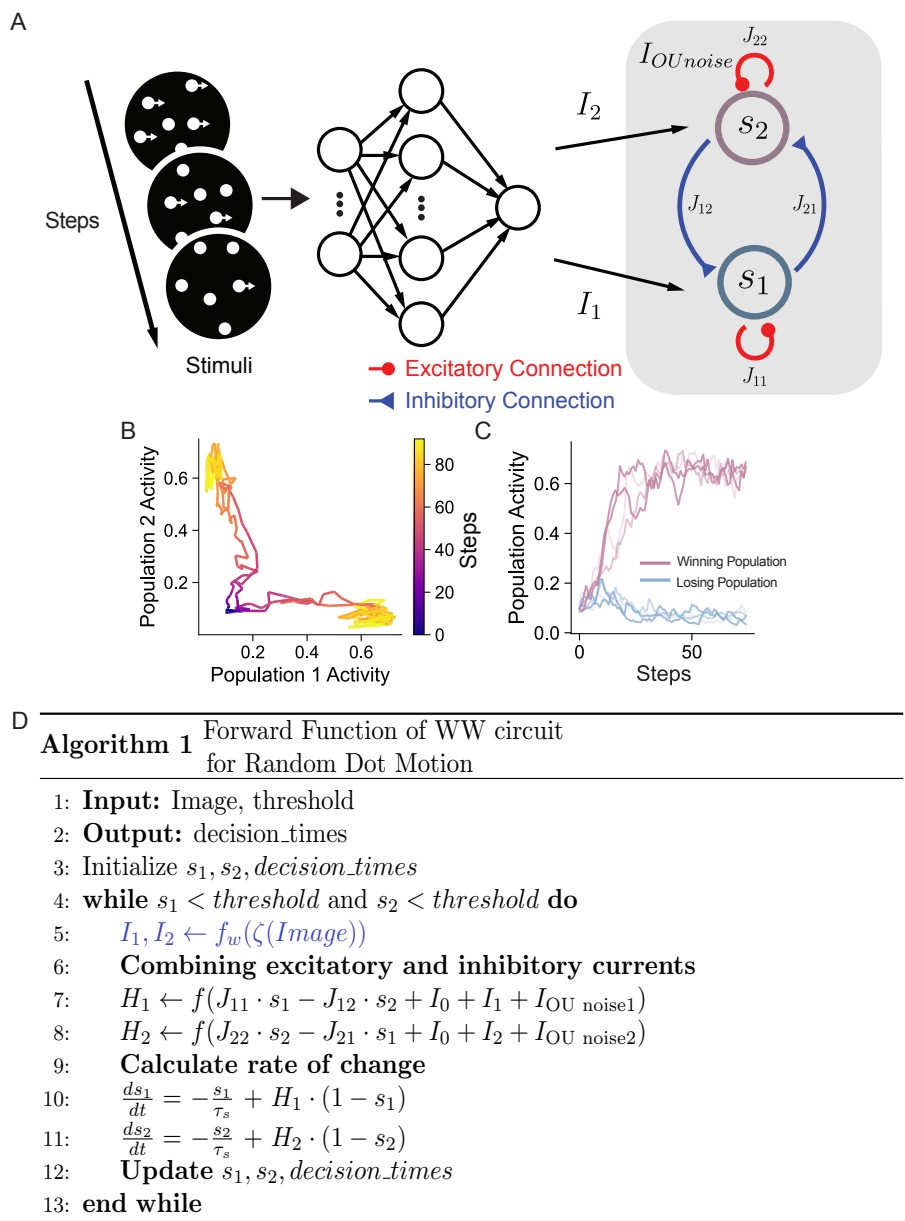

**D**

| **Algorithm 1** | Forward Function of WW circuit for Random Dot Motion |
|---|---|

1: **Input:** Image, threshold
2: **Output:** decision_times
3: Initialize $s_1, s_2, decision\_times$
4: **while** $s_1 < threshold$ and $s_2 < threshold$ **do**
5: $\quad I_1, I_2 \leftarrow f_w(\zeta(Image))$
6: $\quad$ **Combining excitatory and inhibitory currents**
7: $\quad H_1 \leftarrow f(J_{11} \cdot s_1 - J_{12} \cdot s_2 + I_0 + I_1 + I_{\text{OU noise1}})$
8: $\quad H_2 \leftarrow f(J_{22} \cdot s_2 - J_{21} \cdot s_1 + I_0 + I_2 + I_{\text{OU noise2}})$
9: $\quad$ **Calculate rate of change**
10: $\quad \frac{ds_1}{dt} = -\frac{s_1}{\tau_s} + H_1 \cdot (1 - s_1)$
11: $\quad \frac{ds_2}{dt} = -\frac{s_2}{\tau_s} + H_2 \cdot (1 - s_2)$
12: $\quad$ **Update** $s_1, s_2, decision\_times$
13: **end while**
14: **return** decision_times

Figure S2: **Illustration of the RTified WW circuit.** To RTify feedforward neural networks, we used an RNN based on the WW circuit. Here, we consider a binary classification on random moving dots for illustration (**A**). When receiving a visual input, the two populations sensitive to left/right directions compete with each other (**B**) and accumulate evidence until one of them reaches a threshold (**C**). The number of time steps needed for the RNN to reach the threshold is defined as the "model RT" and is used to predict human RT. We provide pseudo-code for a more detailed description of the circuit (**D**). Here, $f(x) = \gamma \cdot max(ax - b, 0)/(1 - exp(-d(ax - b)))$ is a fixed nonlinear function. And the model and equations marked blue are how we extend WW model.

## A.3 Extended results for RDM task

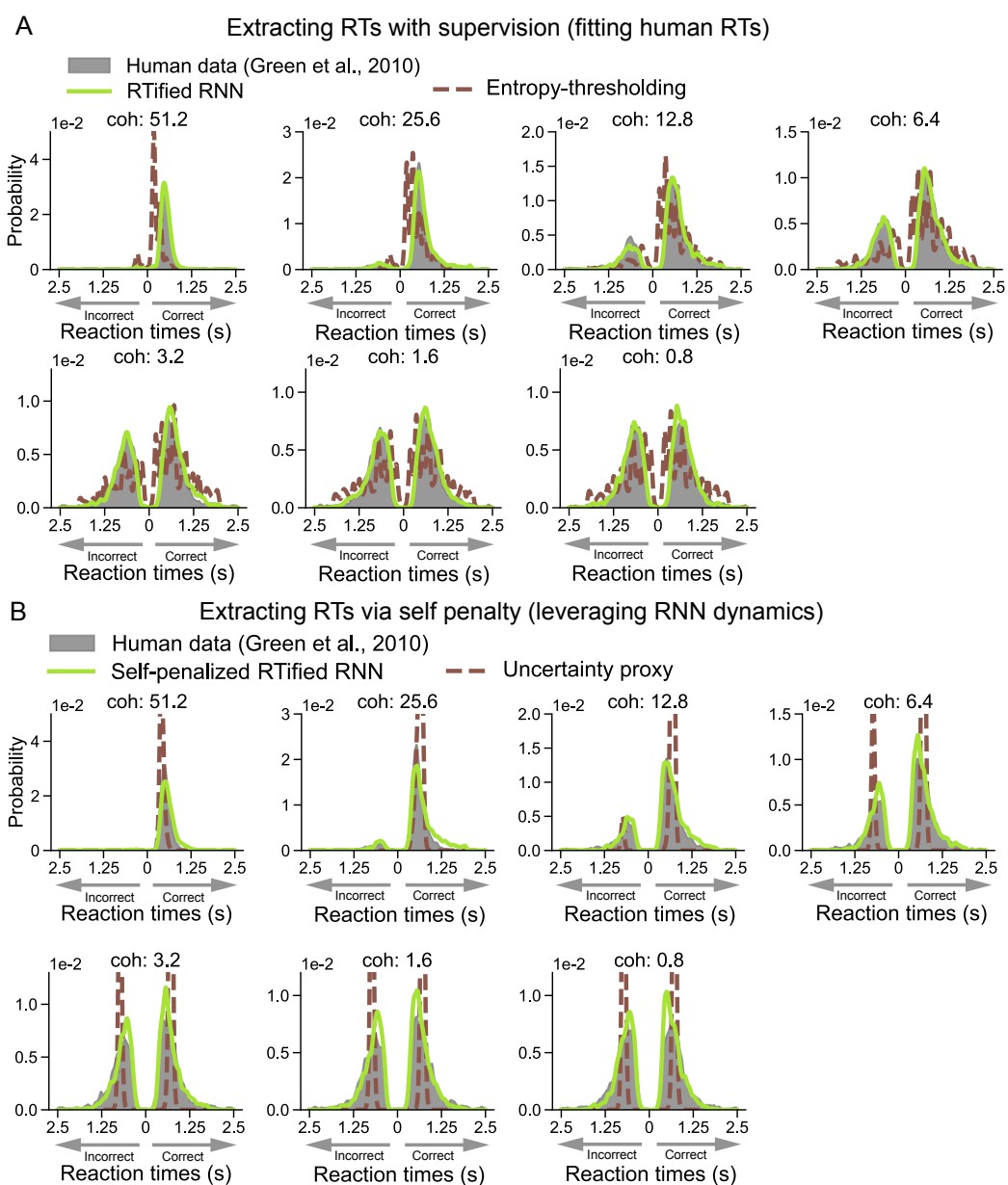

Figure S3: **RTified model evaluation on a RDM task [24] across all coherences.** Human data are shown as a gray shaded area, and model fits are shown for **(A)** the "supervised" setting where human behavioral responses are used to train the models and **(B)** the "self-penalized" setting where no human data is used. Our approach (green) outperforms the two alternative approaches (brown), i.e., entropy-thresholding [29] for the "supervised" and uncertainty proxy [30] for the "self-penalized" settings.

Figure S4: **RTified WW model on a RDM task [24] across all coherences.** We combined our RTified WW module with a 3D CNN to fit human RTs collected in an RDM task [24]. Human data are shown as a gray shaded area and model results are shown as orange solid lines.

## A.4 Additional Experiment for RDM task

Recurrent neural networks generate a sequence of outputs across time steps, raising the question of how to convert these multiple outputs into a single final prediction. In the main text, we combine all the network's outputs up until the decision time point. Here, as an additional experiment, we only use the network's output at the decision time point. In both "supervised" and "self-penalized" settings, our approach surpasses two alternative approaches [29, 30] (see Fig. S5).

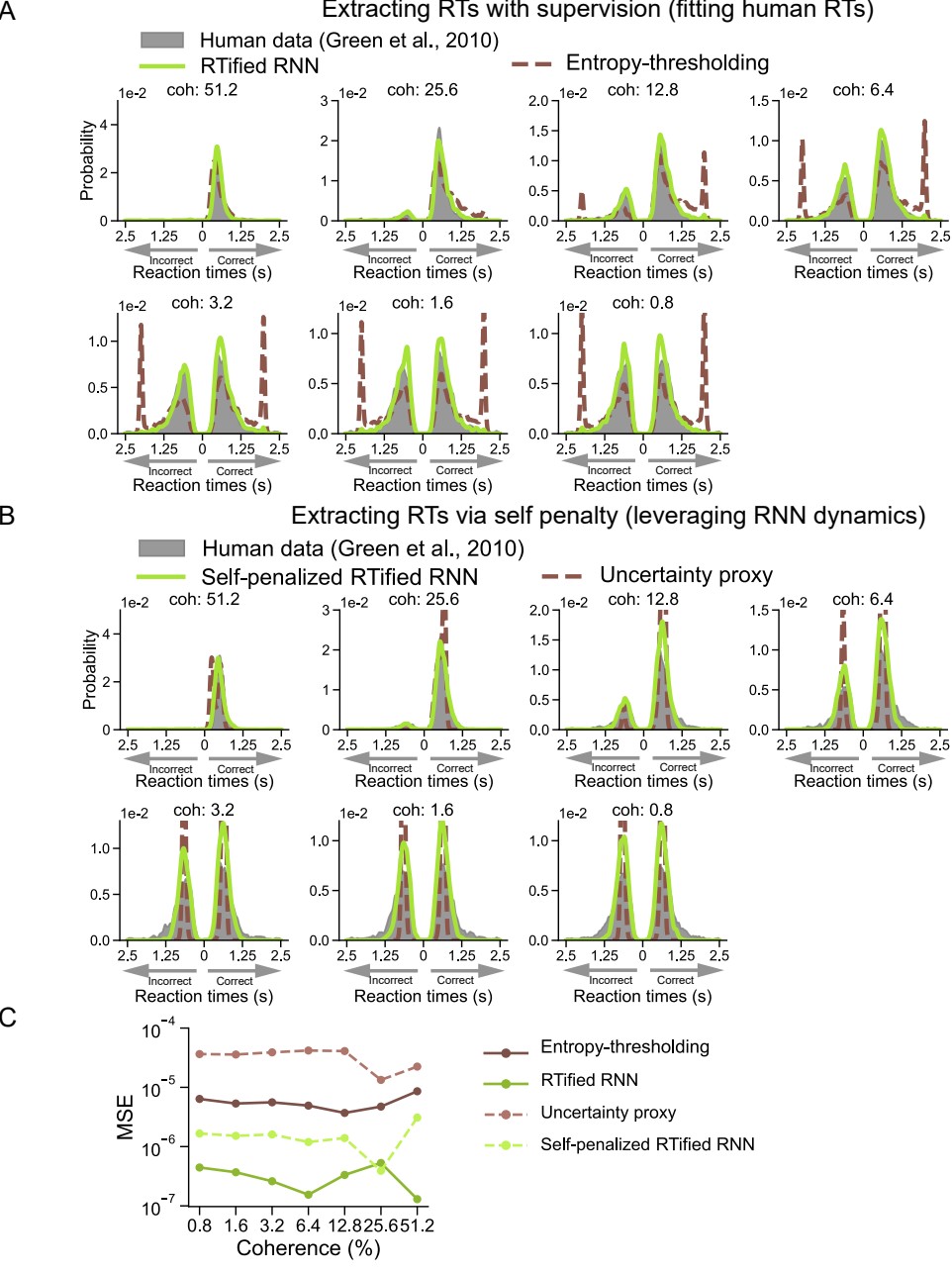

Figure S5: **RTified model evaluation on a RDM task [24] across all coherences.** Human data are shown as a gray shaded area, and model fits are shown for **(A)** the "supervised" setting where human behavioral responses are used to train the models and **(B)** the "self-penalized" setting where no human data is used. Our approach (green) outperforms the two alternative approaches (brown), i.e., entropy-thresholding [29] for the "supervised" and uncertainty proxy [30] for the "self-penalized" settings. MSE comparison is shown in **(C)**.

