# OpenReview forum: "RTify: Aligning Deep Neural Networks with Human Behavioral Decisions"
_NeurIPS.cc/2024/Conference — NeurIPS 2024 poster_

### Official Review · Reviewer_QgJJ · 2024-07-10

**Soundness:** 2
**Presentation:** 2
**Contribution:** 2
**Rating:** 4
**Confidence:** 3

**Summary:**

The  paper proposes a novel framework to align the temporal dynamics of RNNs and human reaction times for visual recognition tasks. The framework starts from a task-optimized RNNs and trains a function $f_w$ to transform the activity (hidden state) into a real-valued evidence measure $e(t)$ (learnable) that will be integrated over time by an evidence accumulator (seems to be a summation). To train the stopping threshold and the function $f_w$, authors propose 2 approaches: supervised and self-supervised.

**Strengths:**

1. The problem sounds interesting - adaptive computation time and its relationship to human reaction times.
2. The solution is novel, using WW models and particular training loss is are not common in NN field.
3. The method, RTify, is a nice way to train with non differentiable loss.

**Weaknesses:**

1. It sounds like the paper writing is a bit overcomplicated. The actual approach is applying RNN (in the WW form that is not clear) on top of the model like CNN, and then learning to reduce the number of RNN iterations via the supervision or via regularization.
2. Benchmarks selected for visual experiments are not strong. Why not to consider CIFAR10/100, ImageNet, COCO detection etc?
3. Details of WW models are not mentioned in the main paper. Probably, listing PyTorch pseudo code for forward pass will be helpful here.
4. The problem of adaptive compute is not novel for RNNs. Please consider comparing with other methods in the filed like ACT ("Adaptive Computation Time for Recurrent Neural Networks" by Alex Graves).

**Questions:**

1. The formulation of $\Phi$ function is not clear, and how the loss is actually propagated in (3) and (4).
2. Figure 3, the Wong-Wang model is mentioned, the structure should be illustrated for the reader to understand it without going to the reference. This is because RTifing is one of the main contributions of the paper.
3. If understand correctly, the dynamic computation time comes only from WW model, and the CNN stays exactly the same, and runs only once. Probably CNN run time dominates the full model, please comment on this.

**Limitations:**

Not listed in the paper as a separate section.

---

> ### Author Rebuttal · Authors · 2024-08-07
>
> **Methods of the current paper and the comparison between ACT**
>
> We will definitely strive to make the paper clearer and simplify our writing. We think our writing may have caused some confusion and here are some points we would like to clarify.
>
> First, our main goal is to align the computational steps of an RNN with human RTs. Our main approach is not simply to place an RNN on top of a convolutional neural network (CNN) but to develop a differentiable framework that allows us to train an integrated model end-to-end on experimental data. Results from Figures 4-6 clearly show that we are able to derive such an RT measure better than previous models. After establishing that RNNs can be aligned with human decisions, we extended our work to align CNNs with human decisions. As this reviewer correctly pointed out, we, therefore, integrated an RNN with a CNN. Combined with our RTify method, we obtained an RT measure directly from the RNN steps required to complete the task. Figure 7 shows results similar to those in Figures 4-6.
>
> Second, we used RTify for a dual purpose: (1) to fit human RTs, and (2) to flexibly minimize the number of computational steps needed to complete the task. This requires a computational method to directly optimize RTs. This cannot be done with ACT because it does not incorporate a way to directly train a loss function with respect to RTs. Instead, ACT can only indirectly be used for (2) by maximizing cumulative activities. Our gradient approach in Section 2.0 and Appendix D allows us to effectively achieve both goals (1) and (2) within the same framework.
>
> We have provided a detailed comparison and clarification in the general rebuttal section to aid understanding. In brief, ACT is only compatible with image tasks using a self-supervision goal. Our results show that the model RT derived from the ACT method does not significantly correlate with human RT (r = 0.05, p = 0.24), see Fig R2.
>
> **Dataset issue**
>
> We understand that a large dataset will help strengthen our work. However, to our knowledge, none of the large datasets mentioned here (CIFAR10/100, ImageNet, COCO detection) come with human behavioral data and, hence, cannot be used to evaluate our model.
>
> **WW architecture**
>
> We added a detailed explanation (see general rebuttal.)
>
> **Formula of Phi**
>
> For the $\Phi$ function, it is a summation of $e(t)$, and $e(t)$ is $f_w (h_t)$. Where $f_w$ is achieved with an MLP.
>
> There are two two different loss formulations for two purposes  in the paper (i.e. fitting human RTs and minimizing time steps.)
>
> For fitting RT, please directly refer to formula (3). Here $F$ is the mean square error(MSE) between the human RT distribution and model RT distribution.
>
> For minimizing time steps, please refer to formula (4). And here we elaborate more on how to get the gradient. There are two terms in formula(4). One is the regular cross-entropy loss, which is totally differentiable and not worth discussion. The other is the penalty of RT, i.e. ($l_y \cdot \Tau_{\tau}$). When we are trying to get the gradient of the penalty term over $w$, since it is bivariate function, we first apply the chain rule $\partial(
>   l_y \cdot \Tau_{\tau})/\partial w = \Tau_{\tau} \cdot \partial l_y / \partial  w + l_y \cdot \partial \Tau_{\tau}/\partial w $.
> (formula *) For the first term of (formula *), since $l_y$ are just the logits of the network, that part is differentiable in the regular way. For the second term of (formula *), we need to apply formula (3) with a special case that $F$ is the identity function.
>
> **Runtime analysis of WW model**
>
> We are not entirely sure we understand the question because runtime is not really relevant here. The RT in the WW model is the number of time steps it takes for the circuit to make a decision. And we use that measure to model human RT.
>
> In case we misunderstood, here is the actual runtime complexity of the WW model:
>
> Input computation (2 matrix operations: one for each neuron).
>
> Synaptic activity update (2 differential equations solved).
>
> Noise update (2 noise generation and update operations).
>
> Trajectory update (storing the values for later use).
>
> Therefore, the WW circuit runtime is primarily governed by the number of time steps and the size of the batch to be processed, which results in a complexity of O(time_steps * batch_size). The reviewer is right that the total runtime of the RTified CNN is dominated by the forward step of the CNN since it contains most of the parameters, but we want to emphasize again that it has nothing to do with the RT.

---

> > ### Comment · Reviewer_QgJJ · 2024-08-13
> > **Update on review**
> >
> > I would like to thank the authors for the rebuttal. Perhaps due to my limited knowledge in the field of human reaction time, I am unable to fully grasp the potential of this paper. My current feedback is that WW is still not clearly explained for practitioners, and providing the source code would help with this. I also don't see a significant breakthrough in the method itself, as it seems to be an RNN on top of features. Therefore, I maintain my score of Borderline Reject; however, I would not oppose the acceptance of this paper given the more positive reviews from others. I suggest that the authors focus on explaining the approach in more detail and ensure that the explanation benefits practitioners who are not experts in the field of reaction time.

---

> ### Author Response · Authors · 2024-08-13
>
> Dear reviewer,
>
> In summary your feedback can be listed in three main points. Let us  respond to each one of them :
>
> 1. *WW is still not clearly explained for practitioners, and providing the source code would help with this*
>
> **We provided the source code in the original submission. See wong_wang.py** in the supplementary.zip,  for details on the implementation of WW. And **we emphasize** that our method is suitable for any reasonable decision-making circuit besides WW, in fact any RNN.
>
> 2. *"I also don't see a significant breakthrough in the method itself, as it seems to be an RNN on top of features"*
>
> We respectfully disagree. Our main approach is not simply to place an RNN on top of features, but to have the **RNN adaptively decide to stop based on the features of the RNN itself**.Our technical breakthrough lies in the differentiability of time steps, which is neither solved in previous adaptive computing methods for RNNs (e.g. ACT), nor solved in human decision models (e.g. DDM).  As we pointed out in our rebuttal. Unlike previous phenomenological models (e.g. DDM), our method allows us to develop image-computable, nonlinear adaptive models of decision making.
>
> 3. *"I suggest that the authors focus on explaining the approach in more detail and ensure that the explanation benefits practitioners who are not experts in the field of reaction time."*
>
> We thank the reviewer for the feedback on explaining the method to a broader audience, which we will take into account for the camera-ready version. In general, however, we believe that this paper provides an important tool for neuroscience researchers to link the phenomenon of reaction times to neural mechanisms of decision making.

---

### Official Review · Reviewer_UxtZ · 2024-07-12

**Soundness:** 3
**Presentation:** 3
**Contribution:** 3
**Rating:** 7
**Confidence:** 4

**Summary:**

Here the authors present a method for fitting neural network outputs to reaction times. The main technical step is setting up the calculation of a reaction time as an accumulation of a decision signal over time steps and creating a differentiable computation of the time when this accumulation crosses a threshold. Here the authors use a simple linear interpolation between the time points to allow reaction times between the time points and a differentiable dependency on the input signal for accumulation. The authors use this possibility to optimise reaction times to match average reaction times of human observers and to minimise reaction time of networks optimised for performance. By using an existing network architecture (“Wong-Wang”) the authors are able to create realistic reaction time distributions as well.

**Strengths:**

The method is successful in fitting Reaction times and is very generally applicable as the only requirement is a scalar input for the preferred decision.
Also the method is well grounded in the literature on how reaction time distributions are generated.
Also the fit is better than two existing competitor methods.

**Weaknesses:**

While the method presented here is fairly general it is still fairly specific in terms of the data it applies to: Timed categorisation tasks without cues, planning, rewards etc. This is a step forward from just accuracy, but clearly far from all decision making behaviour of humans.

While the deep neural network based methods are sensible comparisons, the adapted Wong-Wang architecture received little test in this manuscript. For insights into the modelling, I believe a more thorough discussion of the concrete model used and which aspects of it are relevant for getting the RT distributions right could improve my trust into this method.

The data used here for the natural image categorisation task is not a great choice. While it has a substantial amount of images it is an MTurk dataset with corresponding limitations in accuracy of stimulus presentations and measurements. There are plenty of better controlled psychophysics datasets that could be used for evaluations of RT models.

**Questions:**

Can this approach be extended towards creating predictions for other manipulations of reaction time?

**Limitations:**

While I don’t think the authors are hiding any negative social impacts or severe limitations, I do think they are vastly overestimating the broader impact of a method for predicting RT distributions based on neural networks. I like this work, but it does not allow alignment with the full spectrum of human behavioural data, it will not make DNNs more trustworthy for now and there is exactly one supervised and one less supervised situation, not many.

---

> ### Author Rebuttal · Authors · 2024-08-07
>
> **Psychophysics tasks**
>
> First, for the natural image dataset, we are using the dataset already peer-reviewed and published in Nature Neuroscience (Kar et al., 2019). As the reviewer pointed out, this is done via the Amazon Mechanical Turk (Mturk) platform. The reviewer is correct that these online experiments are not as easy to control compared to in-lab experiments but they come with many benefits (most prominently data collection is orders of magnitude faster to collect compared to in-lab experiments) and with they are now widely used by a large number of psychology labs (see Buhrmester et al., 2018 for a review).
>
> We acknowledge that behavioral choices and RTs provide only a partial picture of human decision-making. However, we believe that these RTs and accuracy are by far the two most crucial metrics in the field of visual decision-making (Ratcliff, 2002; 2008; Chuan-Peng et al., 2002; Fengler et al., 2002), as they often reflect fundamental mechanisms of information processing. We hope our work will inspire experimentalists to conduct both well-controlled in-lab experiments and online experiments to further explore RTs and decision-making behavior.
>
> **WW architecture**
>
> First, we will add a detailed discussion of the details of the WW model in supplementary information (see general rebuttal) and we hope it can clarify the confusion.
>
> Second, we also followed this reviewer’s suggestion and conducted an ablation study of the circuit for the random dot motion tasks. Our findings revealed an intriguing result that may prompt further research: ablating self-excitation causes the model to fail, resulting in flat RT distribution (MSE increases by 3369%), while ablating cross-inhibition deteriorates model performance (MSE increases by 38.5%). We thank the reviewer for bringing this up.
>
> **Predictions for other manipulations of reaction time**
>
> Because RTify tackles the task on a frame-by-frame basis, a practical future experiment could investigate how individual frames of the random moving dots stimulus affect RTs. Typically, in random dot motion tasks, researchers use a consistent coherence level and direction throughout each trial. However, our method enables frame-based interventions. For example, researchers could enhance the coherence of certain frames to determine the importance of early, mid, or late frames. Alternatively, specific frames could be reversed to an opposite direction or masked entirely. RTify opens up these possibilities, offering researchers new avenues for experimentation.
>
> **Lack of limitations and overstating broader impacts**
>
> We will soften our claims regarding broader impacts and include a limitation section so that readers can correctly understand the pros and cons of our current methods.
>
> **Reference:**
>
> Kar, K., Kubilius, J., Schmidt, K., Issa, E. B., & DiCarlo, J. J. (2019). Evidence that recurrent circuits are critical to the ventral stream’s execution of core object recognition behavior. Nature neuroscience, 22(6), 974-983.
>
> Fengler, A., Bera, K., Pedersen, M. L., & Frank, M. J. (2022). Beyond drift diffusion models: Fitting a broad class of decision and reinforcement learning models with HDDM. Journal of cognitive neuroscience, 34(10), 1780-1805.
>
> Buhrmester, M. D., Talaifar, S., & Gosling, S. D. (2018). An evaluation of Amazon’s Mechanical Turk, its rapid rise, and its effective use. Perspectives on psychological science, 13(2), 149-154.
>
> Chuan-Peng, H., Geng, H., Zhang, L., Fengler, A., Frank, M., & ZHANG, R. Y. (2022). A Hitchhiker’s Guide to Bayesian Hierarchical Drift-diffusion Modeling with Dockerhddm.

---

> > ### Comment · Reviewer_UxtZ · 2024-08-12
> >
> > I read the author rebuttals and came to the conclusion that my initial assessment was fair. This provides a nice solution to the relatively narrow field of using human RTs for fitting or evaluation.
> >
> > Actually reducing the overstatements will be appreciated!

---

### Official Review · Reviewer_imBE · 2024-07-13

**Soundness:** 3
**Presentation:** 3
**Contribution:** 3
**Rating:** 6
**Confidence:** 3

**Summary:**

The paper introduces a new framework for training vision systems using human reaction times. The approach allows for dynamic integration of visual reasoning with decision making by incorporating human behavior over time. The approach is shown to be beneficial over a range of psychophysics tasks.

**Strengths:**

* A novel incorporation of new human behavior data, reaction time. The authors correctly point out a gap in incorporating this sort of behavior into models.
* A novel and intuitive framework for incorporating reaction time in a non-linear method. The evidence accumulator was intuitive and well-explained.
* Interesting results that demonstrate the RTify can train models that match human reaction times.
* Application to both simulated (random dot) and real (object detection) visual tasks.

**Weaknesses:**

* The paper doesn’t explain why human RTs are useful. I read the introduction a few times and couldn’t explain the intuition about what incorporating RTs is solving. At some level, I believe there is an intuitive explanation: reaction times give a distribution of responses over time and therefore, this measure directly incorporates decision-making (a process of time) and visual reasoning at each time step.
* A more robust explanation of human RT would be extremely helpful to explain more of the deep intuition. This might be entirely obvious but the paper makes RT seem like a novel contribution without explaining why it might be important.
* It was a bit hard to connect your results to the reported performance of the models. Could results like ones reported in line 277 be put in a table?
* (Nit) I found figure 7a a bit hard to understand. What do the correct and incorrect arrows mean?

**Questions:**

* I was wondering how RT impacted performance for a while. Was there a good sense of this? Maybe the presentation could be cleaned up to present this more systematically.
* I wasn’t able to get a sense of runtime for the WW circuit. This may also need discussion.

**Limitations:**

* I feel like some limitations were missing. For example, even though you don’t theoretically need human data in the self-supervised set-up, from my perspective, there is some reliance on human data for validation of this approach. Does this bottleneck the amount of data you can use?

---

> ### Author Rebuttal · Authors · 2024-08-07
>
> **Why is modeling RT important?**
>
> Modeling RT is crucial for two main reasons. First, because of the so-called speed-accuracy trade-off (fast decisions come with high error rates, and vice versa), a model that explicitly accounts for behavioral decisions and RTs is expected to reflect human brain processing better.
>
> Second, RT itself is an important measure reflecting the processing speed and efficiency of cognitive functions (Heitz, 2014) . It captures the dynamic nature of the visual and decision-making systems (Kar et al., 2019; Wyatte et al., 2014; Ratcliff 2002; 2008). Numerous studies have used RT as a behavioral measure to understand how stimulus complexity, number of potential responses, or stimulus-response compatibility affect human decision making (Kaswan and Young, 1965; Fitts and Seeger, 1953; Fischman, 1984; Reddi and Carpenter, 2000). Integrating mechanisms responsible for behavioral choices and RTs will thus lead to a more complete model of human visual processing.
>
> **Organizing the model performance results**
>
> Thank you for the suggestion. We have organized the model results into a table (see rebuttal pdf).
>
> **Figure 7a**
>
> The distribution shown in Figure 7a is a distribution of signed RT, where the RT of a correct trial is positive, while the RT of an incorrect trial is negative. The distribution of signed RT is routinely used in decision-making studies to summarize experimental data into a single plot: (1) it shows the distribution for RT in all trials (2) the relative proportion of positive RTs over negative RTs reflects the accuracy. In this case, when our model shows a close match with such a distribution, we are successful in modeling both the accuracy and RT at the same time.
>
> **How does RT impact performance?**
>
> We are not entirely sure whether the reviewer is referring to the performance of human participants or the RTify model.
>
> In terms of human performance:  RT and performance are interdependent, as both behavioral choices and RTs are functions of the stimuli. Recent studies have found that if one is prone to error, one may deliberately take more time to decide (Adkins et al., 2024). This example shows how two factors may dynamically regulate each other. We believe this feature makes our RTify an important method since it now helps explain both the choices and the RTs.
>
> In terms of the performance of how RTify predicts human decisions: this is a great question and we would love to look into this more closely. We expect a model trained on combined choice and RT data to better predict choice data. This could in theory be easily tested. Unfortunately, the datasets we use either provide choice data without the stimuli (Green et al., 2010) or they do provide the stimuli but do not provide choice data (Kar et al., 2019). We hope our paper can encourage researchers to collect more comprehensive datasets in order to better validate our method.
>
> **Runtime of the WW circuit**
>
> Thank you for raising this. We do see that more discussion over the WW circuit will help readers better understand the application of RTify over a specific canonical circuit. To this end, we put a detailed discussion of WW circuits in the general rebuttal section.
>
> **Limitations missing**
>
> We apologize for not being explicit about the limitations. We carefully took your suggestion into consideration and have written a separate limitation section (see general rebuttal section.)
>
> **Reference:**
>
> Kar, K., Kubilius, J., Schmidt, K., Issa, E. B., & DiCarlo, J. J. (2019). Evidence that recurrent circuits are critical to the ventral stream’s execution of core object recognition behavior. Nature neuroscience, 22(6), 974-983.
>
> Green, C. S., Pouget, A., & Bavelier, D. (2010). Improved probabilistic inference as a general learning mechanism with action video games. Current biology, 20(17), 1573-1579.
>
> Heitz, R. P. (2014). The speed-accuracy tradeoff: history, physiology, methodology, and behavior. Frontiers in neuroscience, 8, 150.
>
> Kar, K., Kubilius, J., Schmidt, K., Issa, E. B., & DiCarlo, J. J. (2019). Evidence that recurrent circuits are critical to the ventral stream’s execution of core object recognition behavior. Nature neuroscience, 22(6), 974-983.
>
> Wyatte, D., Jilk, D. J., & O'Reilly, R. C. (2014). Early recurrent feedback facilitates visual object recognition under challenging conditions. Frontiers in psychology, 5, 674.
>
> Ratcliff, R., & Tuerlinckx, F. (2002). Estimating parameters of the diffusion model: Approaches to dealing with contaminant reaction times and parameter variability. Psychonomic bulletin & review, 9(3), 438-481.
>
> Ratcliff, R., & McKoon, G. (2008). The diffusion decision model: theory and data for two-choice decision tasks. Neural computation, 20(4), 873-922.
>
> Kaswan, J., & Young, S. (1965). Effect of stimulus variables on choice reaction times and thresholds. Journal of Experimental Psychology, 69(5), 511.
>
> Fischman, M. G. (1984). Programming time as a function of number of movement parts and changes in movement direction. Journal of Motor Behavior, 16(4), 405-423.
>
> Reddi, B. A. J., & Carpenter, R. H. (2000). The influence of urgency on decision time. Nature neuroscience, 3(8), 827-830.
>
> Adkins, T. J., Zhang, H., & Lee, T. G. (2024). People are more error-prone after committing an error. Nature Communications, 15(1), 6422.

---

> > ### Comment · Reviewer_imBE · 2024-08-12
> >
> > Thank you for the careful response. I have read it over and believe all my concerns are addressed. My familiarity with psychophysics tasks is a bit low so I will keep my score as is to reflect my lack of familiarity with the domain.

---

### Official Review · Reviewer_6b9L · 2024-07-15

**Soundness:** 3
**Presentation:** 4
**Contribution:** 3
**Rating:** 7
**Confidence:** 5

**Summary:**

The authors present RTify, a novel approach that leverages Recurrent Neural Networks (RNNs) to model decision response times. RTify offers a dual benefit: it can align human and RNN response times, and it can self-supervise RNNs to optimize the speed-accuracy tradeoff. Through evaluations on both synthetic and natural image recognition tasks, the authors demonstrate that RTify outperforms existing methods (Goetschalckx et al., 2024) in fitting human response times, achieving superior results in both supervised and unsupervised settings.

**Strengths:**

Strengths:
- The authors provide a compelling motivation for modeling human response time behavior using RNNs, addressing a significant gap in prior large scale vision models (used in machine learning) that often neglect temporal dynamics of decision-making. This work effectively bridges that gap, offering a more comprehensive approach to modeling human response behavior using RTified deep neural networks.
- A significant advancement over prior work is the proposed method's ability to learn the stopping criterion end-to-end, either with human RT supervision or by optimizing the speed-accuracy tradeoff. This departure from relying on human-defined thresholds for stopping the RNN's processing marks a substantial improvement, enabling a more adaptive and autonomous modeling of reaction times.
- The paper is well-written, with articulate language and high-quality figures that clearly elucidate the proposed method.
- The experimental findings are intriguing, and the comparative analysis with Goetschalckx et al., 2024 is particularly informative, highlighting the notable improvements introduced by the current technique.

**Weaknesses:**

Weaknesses:
- Although the proposed RTify method is a significant extension of prior art in modeling human response times, it bears notable similarities with existing approaches, such as Adaptive Computation Time [1] and its adaptation to ConvRNNs, AdRNNs in [2]. A more explicit discussion of these similarities and differences would strengthen the paper. Specifically, the analogy between [1, 2]'s halting probability ($p_t$, $P_t$) and RTify's evidence accumulation ($e(t)$, $\phi(t)$), as well as the resemblance between the self-penalized loss function in Eq. 4 and the objective functions in [1] and [2], warrants clarification. The key distinction between RTify and ACT appears to be the learnability of $\tau$ in RTify, whereas ACT treats this threshold as a hyperparameter (to clarify, I find closing this gap to be a strong contribution); a clear discussion of this difference (and any others that I have missed) would be beneficial for the readers.
- In lines 67-69, the authors argue that prior work fails to provide a mechanistic account of the underlying decision processes governing temporal dynamics. However, the current work I believe also suffers from this limitation. This paper may further benefit from a more detailed mechanistic explanation of the temporal dynamics of decision-making in RTify. While the paper demonstrates impressive empirical results, a deeper understanding of the underlying mechanisms would further enhance its impact.
- Minor: please check for spelling errors (e.g. add subscript t to $\phi$ in Figure 1, "Neural Netwok" -> "Neural Network" in same figure etc.). Changing the variable name for halting step $\Tau_\tau$ would improve readability as the threshold for evidence accumulation ($\tau$) also uses the same alphabet.

References:
1. Graves, A. (2016). Adaptive computation time for recurrent neural networks. arXiv preprint arXiv:1603.08983.
2. Veerabadran, V., Ravishankar, S., Tang, Y., Raina, R., & de Sa, V. (2024). Adaptive recurrent vision performs zero-shot computation scaling to unseen difficulty levels. Advances in Neural Information Processing Systems, 36.

**Questions:**

NA. Please refer to my review above.

**Limitations:**

Yes, the authors have sufficiently discussed limitations of their proposed work.

---

> ### Author Rebuttal · Authors · 2024-08-07
>
> **Comparison between ACT**
>
> We will add a comparison in the main text (see general rebuttal section).
>
> **Mechanistic account of RT**
>
> We will revise our wording to provide a more accurate claim about the scope of our work. Here, we emphasize how RTify helps our understanding of RTs. First, our results provide computational evidence that reaction times (RTs) are generated through an evidence accumulation process. Second, our findings provide further evidence that the computational principle underlying RTs is to achieve an optimal speed-accuracy trade-off. Third, RTify offers a tool to model other factors, such as learning and adaptation, that are known to affect RTs. Researchers can apply RTify to various experimental conditions and compare the resulting parameters.
>
> **Spelling errors**
>
> We will fix all these spelling mistakes in the camera-ready version.

---

### Author Rebuttal · Authors · 2024-08-07

We want to thank all the reviewers for providing valuable feedback. We appreciate the time and expertise shared with us, and we are confident that we have addressed all raised concerns. Our paper is now stronger than before. The converging concerns are listed and answered in our general rebuttal, while each unique comment is answered individually.

**What is the difference between our method (RTify) and the existing adaptive computing method in RNNs (e.g. adaptive computing time, ACT)?**

There are three main points we want to mention:

First, the time steps in ACT are not differentiable and thus ACT cannot be used to fit human RTs where the loss is a function of time steps. Therefore, ACT cannot be used in our supervision case.

Second, ACT cannot be applied to image sequences (videos). There are two time components in ACT: the number of recursive steps for processing each frame (N or ponder time), and the number of frames an RNN ‘watches’ (t). ACT optimizes the former but not the latter, which is suitable for minimizing computational resources but not for modeling RT data.

Third, although ACT can technically be run static images (i.e., a sequence of length 1 frame; the ponder time would then be treated as a reaction time) in the self-supervision case, our experiments show that RTs derived from the ACT method do not significantly correlate with human RTs (r = 0.05, p = 0.24) in the image classification task, see Fig. R2.

**What is the specific architecture of the Wong Wang (WW) model?**

Unfortunately, we had to omit details in the original submission due to space constraints. However, we agree with the reviewer and we will provide the missing details. In particular, we have created Figure R1 (please find it in the attached pdf), which will be included in the revised manuscript, along with pseudocode illustrating the model's forward pass.
In our extension of the WW model, we replaced the linear transformation of the coherence levels with a feedforward network. Additionally, we extend the model from 2 neural populations to M neural populations (We put the figure for M neural populations in Figure 3 and also a figure for 2 neural populations in Figure R1). Following these modifications, the parameter count in the WW circuit increases to O(M^2). As mentioned in the main text, the original WW model's manual tuning is practical for small M but infeasible as M grows. With RTify, the entire architecture is trained with backpropagation, as the ordinary differential equations (ODEs) in the WW model can be transformed into a discrete-time recurrent neural network (RNN), making it compatible with RTify.
The results using the WW model are summarized in Figure 7 in the main paper. It is important to note that we only used the WW model together with convolutional neural networks (CNNs) as opposed to RNNs. We selected the Wong-Wang model because it is the leading neural circuit model of decision making.

**What are the limitations of the current paper?**

Reviewers imBE, UxTz, and Qgjj pointed out that we did not clearly state the limitations of the current paper. In order to make readers understand our method better, we provide a separate limitation section here. And surely we will include this in our camera-ready version.
In our paper, we have described a computational framework (RTify) to align RNNs with human behavioral decisions, including both choices and RTs. RTify offers two benefits: it can be used to align RNN steps with human RTs, and it can be used as a self-supervised approach to train RNNs to learn to find an optimal speed-accuracy tradeoff. However, there are certain limitations that need to be addressed in future work.
First, most of the human datasets used in our study remain relatively small-scale and largely limited to synthetic images because most current benchmarks focus exclusively on behavioral choices and lack RT data. To properly evaluate our approach and that of others, larger behavioral datasets using more realistic visual stimuli will be needed. We hope that this work will encourage researchers to collect novel internet-scale benchmarks that include not just behavioral choices but also RTs.
Second, although behavioral choices and RTs are widely studied in the field of visual decision-making, they only partially reflect the nature of human decision-making. For a more complete model of decision making, non-visual factors including confidence and specific aspects of cognitive strategies (e.g. search) will need to be integrated into the model. We recognize that incorporating all known factors into deep neural network models will be crucial for building a trustworthy AI. Nevertheless, we feel that RTify as a method for modeling both behavioral choices and RTs is a positive first step in this direction.

---

### Decision · Program_Chairs · 2024-09-25

**Decision:**

Accept (poster)

**Comment:**

This submission addresses the challenge of modeling human reaction times (RTs) using recurrent evidence accumulation. The main contribution of this work is a differentiable approximation of simulated RTs. This approximation enables the incorporation of RTs, operationalized as the number of recurrent computation steps until an adaptive stopping criterion is met, into the training loss. This approach is employed to train networks either to fit human RTs or to reach decisions as rapidly as possible. In the latter application—training networks to reach decisions quickly—the authors demonstrate improvements over state-of-the-art methods. To the best of my knowledge, the former application—directly fitting human RTs—is novel.

Most of the reviewers believe that the contribution of this work warrants publication at NeurIPS, and I share their assessment. However, the reviewers commonly pointed out several weaknesses: overreaching conclusions, insufficient discussion of limitations, and a lack of clarity in some technical sections, particularly regarding the Wong-Wang RNN models. The authors have committed to addressing these issues in the revised manuscript. It is crucial that the authors provide well-documented and functional public code. Another point to consider in the revision is the omission of a relevant reference: Rafiei et al. (2024), published online shortly before the submission deadline:

Rafiei F, Shekhar M, Rahnev D. The neural network RTNet exhibits the signatures of human perceptual decision-making. Nature Human Behaviour. 2024 Jul 12:1-9.